# 3D-ZnO Superstructure Decorated with Carbon-Based Material for Efficient Photoelectrochemical Water-Splitting under Visible-Light Irradiation

**DOI:** 10.3390/nano13081380

**Published:** 2023-04-16

**Authors:** Uji Pratomo, Rifky Adhia Pratama, Irkham Irkham, Allyn Pramudya Sulaeman, Jacob Yan Mulyana, Indah Primadona

**Affiliations:** 1Department of Chemistry, Faculty of Mathematics and Natural Sciences, Universitas Padjadjaran, Jl. Raya Bandung Sumedang Km.21, Kabupaten Sumedang 45363, Indonesia; u.pratomo@unpad.ac.id (U.P.); rapadhiaa30@gmail.com (R.A.P.); irkham@unpad.ac.id (I.I.); allyn@unpad.ac.id (A.P.S.); 2Department of Applied Chemistry, Graduate School of Urban Environmental Sciences, Tokyo Metropolitan University, 1-1 Minamiosawa, Hachioji, Tokyo 192-0397, Japan; jacob.mulyana@deakin.edu.au; 3School of Education, Faculty of Arts and Education, Deakin University, 221 Burwood Hwy, Burwood, VIC 3125, Australia; 4Research Center for Advanced Material, National Research and Innovation Agency, Kawasan Sains dan Teknologi BJ. Habibie, Tangerang Selatan 15314, Indonesia; 5Collaboration Research Center for Advanced Energy Materials, National Research and Innovation Agency-Institut Teknologi Bandung, Bandung 40132, Indonesia

**Keywords:** ZnO superstructures, water-splitting, photoelectrochemical, carbon-based material

## Abstract

The depletion of fossil fuels is a worldwide problem that has led to the discovery of alternative energy sources. Solar energy is the focus of numerous studies due to its huge potential power and environmentally friendly nature. Furthermore, one such area of study is the production of hydrogen energy by engaging photocatalysts using the photoelectrochemical (PEC) method. 3-D ZnO superstructures are extensively explored, showing high solar light-harvesting efficiency, more reaction sites, great electron transportation, and low electron-hole recombination. However, further development requires the consideration of several aspects, including the morphological effects of 3D-ZnO on water-splitting performance. This study reviewed various 3D-ZnO superstructures fabricated through different synthesis methods and crystal growth modifiers, as well as their advantages and limitations. Additionally, a recent modification by carbon-based material for enhanced water-splitting efficiency has been discussed. Finally, the review provides some challenging issues and future perspectives on the improvement of vectorial charge carrier migration and separation between ZnO as well as carbon-based material, using rare earth metals, which appears to be exciting for water-splitting.

## 1. Introduction

Water-splitting, which involves separating oxygen and hydrogen from water, is a fascinating approach to the renewable, environmentally friendly, non-polluting, and highly efficient production of hydrogen energy [1,2,3]. Compared to other methods, such as radiolysis [4], thermal decomposition [5], photobiological [6], and photocatalytic [7], photoelectrochemical or PEC is considered simple, cheap, and generates a high solar-to-hydrogen (STH) conversion efficiency [2,8,9]. Figure 1 presents the PEC water-splitting process, in which the core principle is to apply an external bias to a photovoltaic material immersed in an electrolyte solution containing a semiconductor material as a working electrode and a good electrical conductor material as a counter electrode to convert solar energy into hydrogen through a redox reaction process [1,10].

PEC water-splitting involves three main processes: (I) excitation of a semiconductor’s electron as a result of its interaction with photon energy from the light absorption process, (II) separation and transportation of photogenerated electron-hole pairs, and (III) a redox reaction [9]. Considering the energy requirements for the water-splitting process, there are two critical criteria for semiconductors that can be used as photocatalysts in the PEC system [11]. The first criterion is that the semiconductor’s minimum conduction band (CB) value should be more negative than the reduction potential of H^+^/H_2_. Secondly, the minimum valence band (VB) value should be more positive than the oxidation potential H_2_O/O_2_ [12]. Therefore, a bandgap of about 1.23 eV to 3.1 eV (vs. NHE) is needed to provide a significant enough overpotential for the external bias of water-splitting [11,13].

Pan et al. identified several semiconductors that meet the criteria for providing a redox reaction in water-splitting process, and they included Fe_2_O_3_, MoS_2_, TiO_2_, TaON, SiC, CdS, SrTiO_3_, CdSe, and ZnO [14]. Among these semiconductors, ZnO was considered a significant material for PEC water-splitting due to its high electrochemical stability, high electron mobility, abundance in nature, low cost, and non-toxic properties [15]. Despite these advantages, the performance of 1D-ZnO was low because of its rapid electron–hole recombination, limited specific surface area for mass transfer, and insufficient light absorption [16,17,18,19]. Conversely, 3D-ZnO superstructures were regarded as promising material for PEC water-splitting due to their ability to improve light absorption and harvest solar light more efficiently by increasing the quantity of endless light scattering [20,21]. These superstructures were synthesized with various morphologies, such as mulberry-like [22], worm-like [23], wool ball [24], birdcage-like [25], hedgehogs [26], firecracker [27], nanocomb [28], nanoflower [15], and nanoneedle [29].

Carbon-based materials, including carbon nanotubes (CNTs), carbon quantum dots (CQDs), and graphene derivates, such as graphene oxide (GO), reduced graphene oxide (rGO), and graphene quantum dots (GQDs), were used as a viable sensitizer for improving the semiconductor’s photocatalytic efficiency towards broad application [30]. For instance, Olak-Kucharczyk et al. decorated TiO_2_ with rGO to increase its photocatalytic efficiency [31]. Additionally, Wen et al. successfully synthesized urchin-like ZnO/Au/g-C3N4 nanocomposites using the thermal vapor condensation (TVC) technique, which functioned as active and superior photocatalysts for H_2_ production in visible light [32]. The remarkable features of carbon-based material, such as adjustable electrical and optical properties, excellent biocompatibility, and long-term durability, had led to their wide application in nanocomposites material [30,33].

Several outstanding review articles on the use of both 3D-ZnO superstructures and carbon-based material in energy conversion and environmental applications have been published [21,34,35,36]. However, they primarily focus on the synthesis and extensive use of individual 3D-ZnO superstructures and carbon-based materials. Based on available information, there appears to be a considerable lack of reviews that specifically cover 3D-ZnO superstructures/carbon-based material in PEC water-splitting devices. Therefore, this mini review provides updated information about the modification of 3D-ZnO superstructures with carbon-based material for efficient performance. It begins with presenting a summary of the newly developed synthesis strategies and their crystal growth mechanism, followed by the description of several investigation works concerning the morphological effect on water-splitting performance. Furthermore, this review discusses the challenge and possible modification of 3D-ZnO superstructures/carbon-based materials to optimize PEC water-splitting efficiency in visible light range.

## 2. 3D-ZnO Superstructures’ Synthesis and Their PEC Performance

Desai et al. defined superstructures as three-dimensional (3D) geometries composed of controlled morphology and crystal orientation of 1D and 2D nanostructures. However, the critical aspect of 3D-ZnO superstructures’ synthesis technique was the growth of 1D or 2D ZnO on ZnO seeds [21]. Li et al. successfully constructed a 3D-ZnO microsphere using 1D nanorod ZnO as the building block, while Sun et al. branched a ZnO nanowire array to create a 3D-ZnO nanoforest [20,37]. Solution-based or chemical approaches have been commonly utilized to synthesize a wide range of 3D-ZnO superstructures due to their numerous advantages, including low cost, readily adjustable synthesis parameters, and the absence of a requirement for expensive instruments [38]. Co-precipitation, chemical bath deposition, sol-gel, hydrothermal, and electrochemical deposition are some of the solution-based or chemical approaches for synthesizing ZnO [38,39,40,41]. Table 1 presents a concise summary of the advantages and limitations of each solution-based synthesis technique for obtaining a 3D-ZnO superstructure.

Due to the simplicity of the process and equipment, as shown in Figure 2a, as well as its low-cost budget, high yields, and scalability, the hydrothermal technique has been utilized to obtain a 3D-ZnO superstructures [42,43,44]. Ma et al. successfully synthesized a hemispherical 3D-ZnO, as shown in Figure 2b, through a hydrothermal technique, at 150 °C for 12 h [45]. Similarly, Qu et al. employed the same technique to construct hierarchical ZnO flower-shaped microstructures, using sodium hydroxide as an OH-ion source to control the crystal growth mechanism at low temperatures (60 °C) [46]. The experimental parameters commonly used in hydrothermal experiments include reagents, solvents, temperature, and duration [47].

According to some literature in recent decades, the most common method of engineering diverse morphologies of 3D-ZnO superstructures is the addition of a “growth modifier”. This can adjust the rate of nucleation and growth, and provide support or resistance to growth in a certain direction, resulting in face-specific growth. Surfactants, template agents, surface-directing agents, capping agents, or other chemical agents are considered growth modifiers in this context, and are utilized to grow the hierarchy in 3D-ZnO superstructures [49,50]. Several growth modifiers are necessary for either single or mixed forms. Table 2 shows typical types of growth modifiers used to create or adjust various morphologies of 3D-ZnO superstructures.

Capping agents, which are amphiphilic compounds with a polar head group and a non-polar hydrocarbon tail, are crucial stabilizers in colloidal synthesis. This is because they suppress nanoparticle proliferation and prevent aggregation/coagulation. Various types of capping agents such as small ligands, dendrimers, polymers, polysaccharides, and cyclodextrins are used in the synthesis of nanoparticles [50]. In contrast, surfactants serve an important function in suppressing particle agglomeration and flocculation [46]. Chelating agents can also affect particle morphology and grain size by forming a complex structure with the cation. Furthermore, etching agents or etchants are used to dissolve the ZnO surface, resulting in new nucleation sites for continued development, while surface-directing agents drive the crystal formation in a certain direction or orientation [21]. By using such growth modifiers, it is feasible to design a variety of ZnO superstructures with different morphologies. Finally, the subsection below thoroughly explains specific details on various 3D-ZnO crystal growth mechanisms.

### 2.1. Nanoflower-like Morphology

The nanoflower-like morphology is the most common 3D-ZnO superstructure due to its ease of synthesis, polymorphic variety, and potential for a wide range of applications [59]. Growth modifiers are essential for synthesizing nanoflower superstructures, as they restrict or enable the growth of polar planes, produce micelles, and ultimately determine the final morphology. Zhu et al. used CTAB as a surfactant to synthesize various types of nanoflower-like ZnO by adjusting its molar concentration through the hydrothermal method [43]. Furthermore, nanoflowers are essentially a symmetrical arrangement of nanorods, nanoflakes, or even conical-shaped rod-like morphologies. According to Figure 3a, the crystal of the morphology typically grows in three stages, namely seeds/nuclei formation, Zn(OH)_2_ degradation, and self-assembly into a flower-like shape induced by several structural directors.

### 2.2. 3D Wool Ball-like

3D wool ball-like ZnO has the same morphology as wool yarn. As explained previously, most 3D ZnO is constructed hierarchically from its 1D and 2D forms. Septiani et al. built hierarchical ZnO wool ball-like nanostructures from its 2D form (plate-like sub-units of ZnO) by applying glycerol in 2-propanol solvent as a quasi-microemulsion agent [56]. The crystal growth mechanism is depicted in Figure 3b. Furthermore, the effect of reaction time and glycerol concentration on the formation process was investigated. It was observed that the optimum reaction time for assembling the structures was 16 h. The process will not be conducted properly in less than 16 h. Additionally, applying a greater glycerol concentration to the reaction system led to the formation of denser 2D plate-like subunits and well-defined 3D wool ball-like ZnO structures. Shi et al. have also succeeded in producing 3D elliptical wool ball-like ZnO through an alkali-assisted solvothermal method, which differs from the previous method but applies a similar concept of hierarchical construction from 1D nanowire ZnO [60].

### 2.3. Hexagonal Ring-like

In the past three years, Bao et al. reported a 1 min reaction time for manufacturing flower-like and hexagonal ring-like ZnO superstructures [53]. The secret to this achievement was the use of triethylamine (TEA) as a surface-directing agent for blocking the preferential growth of ZnO crystals along the C-axis. The morphology of superstructures varied depending on water content in TEA. At about 10% water content, ZnO crystal nucleation occurs rapidly, resulting in the formation of flower-like ZnO superstructures. Meanwhile, the rugby-shaped structure can only be generated via isotropic growth from center nuclei when the water content is increased from 10% to 50%, due to the low concentration of TEA in mixed solvents. Lowering water content from 10% to 0% led to a low growth rate and the quasi-equilibrium development of ZnO nanocrystals due to the presence of only TEA in the solvent. The unique overdosage alkaline TEA maintained a suitable pH level, causing the hexagonal ring-like ZnO to erode in succession from the center point to the outside, as shown in Figure 3c. The local zinc concentration is highest in the center of the hexagonal ring-like ZnO, resulting in the fastest erosion velocity at the core of the structure. Therefore, hexagonal ring-like ZnO tends to develop when the water content is reduced to 0%, due to the erosion impact of the additional base.

## 3. Morphological Effect of 3D-ZnO Superstructures on PEC Water-Splitting Performances

For several years, one and two-dimensional ZnO has been applied for PEC water-splitting [61]. For example, Liu et al. investigated the performance of various ZnO nanostructures, namely nanorods, nanodisks, nanowires, and nanotubes [62]. The high porosity of nanotubes allows for multiple reflections of sunlight, and their perpendicular alignment provides excellent electron routes, resulting in the greatest PEC performance of any nanostructure. However, the tiny grain size of nanostructures causes varied resistances across surface states and the grain boundaries, leading to impeded charge transfer. 3D-ZnO superstructures have gained attention due to their advantages over 1D and 2D nanostructures. The difference can be observed in their high surface area, porous architectures, and the synergistic interactions of the generated nano-building blocks [63,64,65]. Consequently, ZnO superstructures have superior physical/chemical features, such as greater light-harvesting, improved electrons, more reaction sites, and ion transportation, all of which are critical for several applications [66,67,68].

The surface area of 3D-ZnO superstructures is determined by the construction block dimensions (1D or 2D) and the final arrangement of the assembly. As a result, their morphology is critical to the efficacy of PEC water-splitting. Sun et al. successfully synthesized various morphologies of ZnO nanoforest and compared their PEC performance with nanowires [20]. The highly dense nanoforest ZnO superstructures generate several light-scattering centers, enabling their interaction with every ray of light and resulting in more trapping. According to the spectra in Figure 4a, 3D-ZnO nanoforests (brus-like and willow-like nanoforest) exhibit higher absorption. A study indicated that willow-like superstructures outperform their brush-like counterparts due to a higher aspect ratio. The longer-tuned branches in willow-like superstructures provide more interaction sites, which leads to more light absorption. The LSV of ZnO superstructures in Figure 4b showed a stronger photocurrent than nanowires. The willow-like and brush-like superstructures increased the photocurrent by 126% (current density 0.344 mA/cm^2^) and 267% (current density 0.727 mA/cm^2^), respectively, when compared to nanowires.

Lv et al. reported another exceptional PEC performance of ZnO superstructures by using a hydrothermal method to synthesize a ZnO nanopencil [16]. The STH efficiency of ZnO nanopencil arrays is determined to be around 0.10% (0.88 V vs. RHE), and greater than that of nanowires. This validates the role of the nanopencil structure in boosting the conversion of absorbed light into the photocurrent. The ZnO nanopencil with sharp tips demonstrated good electrical characteristics, owing to its high oxygen-vacancy concentration. Lv et al. confirmed that the existence of oxygen vacancies was an inherent characteristic of ZnO, due to its low formation enthalpy. An increase in oxygen vacancies could prevent electron–hole recombination by trapping photogenerated charges and facilitating the transport of the holes to ZnO–electrolyte interface for the water oxidation process, consequently increasing the photocurrent density, as shown in Figure 4c. In summary, the morphology of both ZnO willow-like nanoforests and nanopencils, as long as they are 3D superstructures, can enhance PEC water-splitting performances by reducing charge recombination through direct conduction pathways along crystalline ZnO superstructures.

## 4. A Recent Modification of 3D-ZnO Superstructures by Carbon-Based Material

The wide bandgap of ZnO in either 0, 1, 2, and 3-dimensional morphology limits their capability to absorb energy, potentially leading to the high absorption of the UV part of solar light [69]. Meanwhile, the solar energy on the Earth’s surface has been divided into three parts. These include 3% of total energy in the ultraviolet (UV) region (300–400 nm), 45% in the visible range (400–700 nm), and the remaining 52% in the near-infrared (NIR) region (700–2500 nm) [70]. One way to overcome this limitation is by sensitizing or composing 3D-ZnO superstructures with a semiconductor that has a lower bandgap. This approach offers an alternative solution to the aforementioned drawback.

The type-II heterostructure of ZnO with carbon-based material has drawn significant attention to PEC water-splitting studies due to its good stability, lower bandgap than ZnO, great performance as sensitizers for improved STH conversion, and ease of modification and production [2,71]. Carbon black (CB), fullerenes (C60), carbon nanohorns (CNHs), graphene (GR), carbon nanodiamonds (CNDs), carbon nanofibers (CNFs), CNTs, and quantum dots (QDs) are examples of carbon-based nanomaterials [72]. Furthermore, they have distinct qualities that are unmatched, leading to extensive use in a wide range of applications.

Tayyebi et al. successfully prepared flower-like carbon-doped ZnO (C-ZnO/rGO) and showed its improved PEC performance [73]. The incorporation of carbon atoms into ZnO micro rods (MRs)’ structure and hybridization with rGO sheets up-shifted the VBM (valence band maximum) by 0.35. Furthermore, the addition of graphene sheets suppressed the decay in photocurrent density observed in ZnO MRs, resulting in a stable photocurrent of up to 100 mA/cm2 at 1.0 V under light irradiation from a 250 W Xe lamp. In another study, Kim et al. reported the enhancement of ZnO current density through a doping strategy with nanocarbon material in ZnO/graphene and ZnO/fullerene composites [74]. The density of ZnO/Graphene and ZnO/Fullerene was improved to 0.15 and 0.235 mA, respectively, under 100 W light irradiation and an applied voltage of 1.23 V. Shim et al. also reported PEC application of ZnO/multilayer graphene (MLG) shells [75]. This showed better efficient electron transfer between ZnO core and the MLG shell, resulting in improved photocatalytic activity and PEC response. The current was improved to 4.3 times that of bare ZnO under an applied potential of 0.2 V vs. Ag/AgCl and 100 mW light irradiation.

Prasadam et al. utilized a single-pot-hybrid chemical vapor deposition (CVD)–atomic layer deposition (ALD) technique to create a composite ZnO core–shell structure with CNTs to prevent ZnO photo-corrosion in aqueous solution [76]. Their CNTs-ZnO core–shell nanocomposite outperforms ZnO thin film covered with nafion, which was previously prepared by Pawar et al., in terms of current density and photostability [77]. The high conductivity of CNTs permits the protected transport of electrons in the CNTs-ZnO core shell, resulting in a maximum current density of 0.55 mA/cm^2^ (1.23 V_RHE_) with photocurrent stability for 8 h. Figure 5a indicates that CNTs play a role as an electron acceptor from the conduction band of ZnO to the π-system of CNTs as a result of the formation of the Schottky junction.

Li et al. enhanced the visible light-driven photocatalytic activity of 3D burger-like ZnO by incorporating CQDs [78]. The electronic transition from HOMO to LUMO level of CQDs under visible light irradiation is attributed to the ease of band gap alignment of CDQs, which is achieved by turning their size and quantum confinement effect. The electrons are transferred to the conduction band of ZnO, as shown in Figure 5c. After modification with CQDs, several improvements were observed in 3D burger-like ZnO, including a six-fold increase in photocurrent response, a decrease in the band gap value from 3.2 eV to 2.82 eV, and a photocatalytic efficiency of 92.48% in 90 min. Using the same strategy, Mahala et al. decorated ZnO nanosheets with CDs to lift the semiconductor’s charge transportation and improve their visible light activity by modifying band alignment in the heterostructure of ZnO/CDs composites, as presented in Figure 5b.

GQDs have been attracting substantial attention as one of the derivatives of CQDs due to their good crystallinity, strong electric conductivity, and rich functional group compared to the other carbon-based material [30]. They have been used by Alam et al. to sensitize their bio-template-assisted hierarchical ZnO superstructures for enhancing the water oxidation kinetics process [79]. The surface area of ZnO superstructures increases from 32 m^2^/g to 45 m^2^/g after modification, resulting in a 77% increase in photocurrent response. These remarkable performances are due to efficient charge separation and reduced electron–hole recombination, which is further proven by the Mott–Schottky analysis result. Table 3 shows a comparative investigation of the sensitization or modification effect of carbon-based materials on 3D-ZnO super-structures’ PEC water-splitting performances.

## 5. Conclusions and Future Perspectives

Understanding of the crystal growth mechanism of various ZnO morphologies is a basic and significant principle for choosing a suitable “growth modifier” for the production of 3D-ZnO superstructures. This minireview describes the crystal growth mechanisms during the synthesis of nanoflower-like, 3D wool-ball-like, and hexagonal ring-like 3D-ZnO superstructures, in addition to providing a discussion of recent synthesis and modification strategies. Among several synthesis methods, hydrothermal is the most feasible, low-cost, and simplest technique for building a 3D-ZnO superstructure. The physicochemical parameters of ZnO superstructures fluctuate according to their morphological shape; thus, their PEC performance differs. Furthermore, the variations in crystal planes, size and aspect ratios, alignment, and surface proved to be significant in determining the performance, even within the same morphological shape.

One of the critical issues that requires attention is that of overcoming the limitation of light absorption in ZnO. Previous studies suggest that sensitizing ZnO with another semiconductor with a smaller bandgap is the most intriguing option to overcome such a drawback. To improve PEC water-splitting reactions, a suitable sensitizer should possess properties such as corrosion-resistance throughout PEC water-splitting reaction, the ability to boost the optical absorption of the semiconductor material, and appropriate band alignment between the sensitizer and the semiconductor to enable suitable separation of the charge carrier [84]. Carbon-based materials such as CNTs, CDs, CQDs, GO, rGO, and GQDs have attracted attention as viable sensitizers to improve the efficiency of PEC water-splitting. Furthermore, the exploration of rare earth metals as an electron mediator to facilitate the Z-scheme PS-C-PS system for improving vectorial charge carrier migration and separation between ZnO as well as carbon-based materials appears to be a promising area for future study. Therefore, it is necessary to conduct a thorough investigation of suitable preparation methods for such materials in PEC applications.

## Figures and Tables

**Figure 1 nanomaterials-13-01380-f001:**
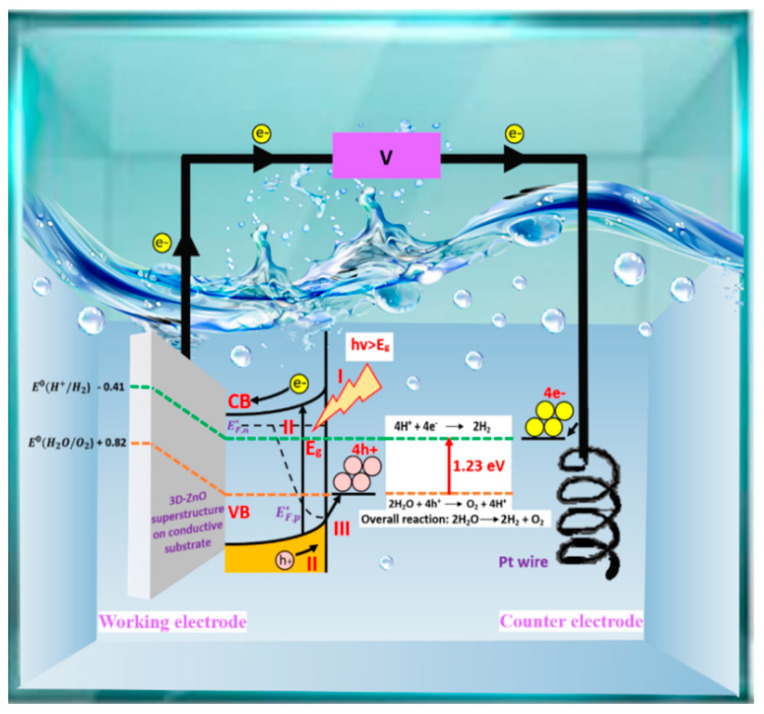
An illustration of the PEC water–splitting process based on an n–type semiconductor photoanode electrically linked to a metal counter electrode under external bias and conducted under alkaline circumstances. The process involves three primary mechanisms: light absorption, charge-carrier separation and transportation, and surface redox reactions.

**Figure 2 nanomaterials-13-01380-f002:**
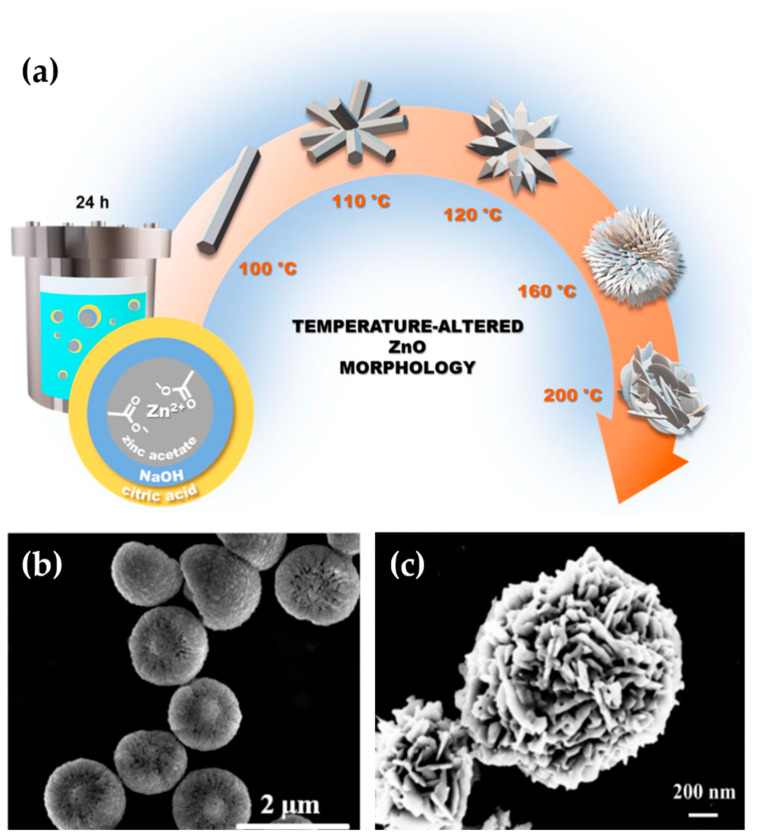
(**a**) Hydrothermal equipment for manufacturing various ZnO superstructures at different temperatures. This was reprinted with permission from [48] and copyright belonging to 2023 MDPI. The SEM image of the hemispherical 3D-ZnO superstructures is shown in (**b**) and has been reprinted with permission from [45]. Copyright 2019 Elsevier. (**c**) Micro-flower-like ZnO superstructures, reprinted with permission from [46], with copyright belonging to 2018 Elsevier.

**Figure 3 nanomaterials-13-01380-f003:**
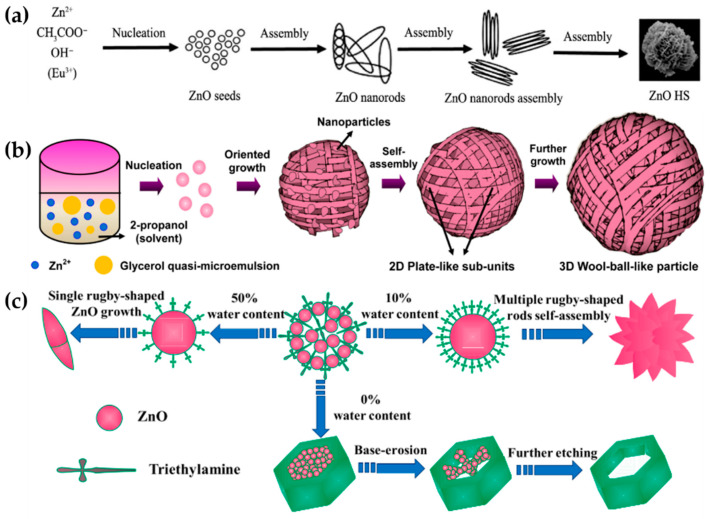
Crystal growth mechanism of (**a**) Nanoflower-like ZnO. Reprinted with permission from [52]. Copyright 2019 Elsevier. (**b**) 3D wool ball-like ZnO. Reprinted with permission from [56]. Copyright 2018 Elsevier. (**c**) Hexagonal ring-like ZnO. Reprinted with permission from [53]. Copyright 19 Elsevier.

**Figure 4 nanomaterials-13-01380-f004:**
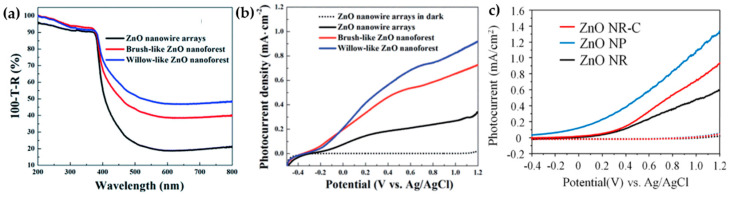
(**a**) Absorption spectra, and (**b**) linear sweep voltammograms (LSV) curve of the nanorod, nanoforest, and willow–like ZnO superstructures. Reprinted with permission from [20]. Copyright 2014 Royal Society of Chemistry. (**c**) Photocurrent density measurement of ZnO nanorod and nanopencil. Reprinted with permission from [16]. Copyright 2014 Elsevier.

**Figure 5 nanomaterials-13-01380-f005:**
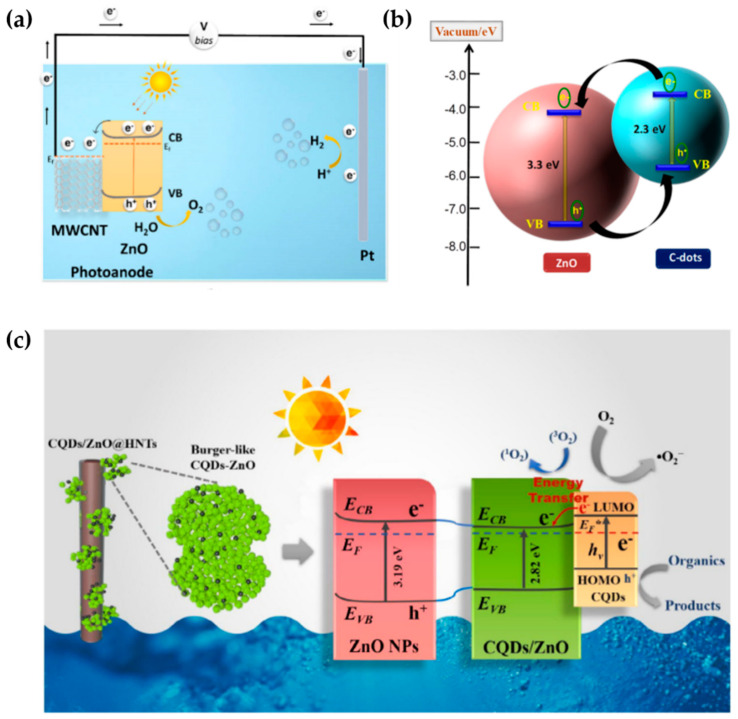
Schematic mechanism illustration of the band configuration and the charge separation during the photocatalytic process in (**a**) CNTs/core–shell ZnO. Reprinted with permission from [76]. Copyright 2022 MDPI. (**b**) CDs/nanosheets ZnO. Reprinted with permission from [2]. Copyright 2020 American Chemical Society. (**c**) CQDs/3D burger–like ZnO. Reprinted with permission from [78]. Copyright 2019 Elsevier.

**Table 1 nanomaterials-13-01380-t001:** The comparison of solution-based approaches for the synthesis of 3D-ZnO superstructure.

Method	Advantages	Limitations
Chemical bath deposition	Various substrates can be used to deposit samples and low cost	Solution waste is produced after each deposition
Electrochemical deposition	Easy to control the morphology and structure by adjusting electrochemical parameters	The growth substrate should be conductive
Hydrothermal	Low cost, green synthesis, simple equipment, and homogeneous production across a vast region	Higher temperature and reaction pressure
Co-precipitation	Rapid and low-cost	The rapid process results in the simultaneous nucleation and growth of ZnO, making it difficult to understand the specific growth process
Sol-gel	Mild synthesis condition, not expensive, and simple	Needs additional purification due to impurities from sol-gel matrix components

**Table 2 nanomaterials-13-01380-t002:** List of different growth modifiers used for the synthesis of 3D-ZnO superstructures.

Growth Modifier	Role of Growth Modifier	Resultant Morphology	Ref.
Ammonia	OH^−^ ions source and etching agent	Nanoforest	[51]
Trisodium citrate dihydrate	Directing agent	Nanoflower	[52]
Triethylamine	Directing agent	Hexagonal ring-like and flower-like	[53]
Sodium dodecyl sulfate	Capping agent	Plate-like	[54]
Polygalacturonic acid	Templating and directing agent	Multi-cage like	[44]
CTAB	Surfactant	Flower-like	[43]
Heparin	Biotemplate and chelating agent	Quasi-microsphere and twinned donut-like hemispheres	[55]
Glycol	Directing agent	Hemi-spherical	[45]
Glycerol	Crystal growth director	Wool-ball-like	[56]
*Cinnamon champora* leaf	Bio-template	Spherical-like	[57]
*Eryngium foetidum* L.	Bio-template	Spherical-like	[58]

**Table 3 nanomaterials-13-01380-t003:** PEC performance data of carbon-based material modified 3D-ZnO superstructures.

Material	Electrolyte & PEC Conditions	Results	Ref.
rGO-modified 3D-ZnO hollow microsphere	Na_2_SO_3_ (0.25 M)-Na_2_S (0.35 M), I = 100 mW/cm^2^ from 300 W xenon lamp	j = 0.1 mA/cm^2^ at 1 V vs. Ag/AgCl (2.7 times better than bare ZnO)	[80]
GO/ZnO flower-like hybrid composites	Na_2_SO_4_ (0.1 M), I = 100 mW/cm^2^ from 300 W xenon lamp	j = 0.09 mA/cm^2^ at 0.65 V vs. RHE (8 times better than bare ZnO)	[81]
g-C_3_N_4_ QDs-decorated ZnO nanosheets	Na_2_SO_4_ (0.5 M), I = 100 mW/cm^2^ from 300 W xenon lamp	j = 1.68 mA/cm^2^ at 1.2 V vs. RHE (1.3 times better than bare ZnO)	[82]
CNTs/ZnO core–shell nanocomposites	NaOH (0.1 M), I = 100 mW/cm^2^ from 300 W xenon lamp	j = 0.55 mA/cm2 at 1.23 V vs. RHE (458% better than bare ZnO)	[76]
GO-modified ZnO triangles	NaOH (1.0 M), UV light ~360 nm	j = 1.52 mA/cm^2^ at 1.45 V vs. RHE (2.08 times better than bare ZnO)	[83]
GQDs/bio-template ZnO superstructures composites	NaOH (1.0 M), I = 100 mW/cm^2^ from 300 W tungsten halogen lamp	j = 0.61 mA/cm^2^ at 1.23 V vs. RHE (77% better than bare ZnO)	[79]
rGO/carbon-doped flower-like ZnO MRs	Na_2_SO_4_ (0.1 M), A 250 W Xe lamp (Oriel) with a 420 nm cut-off filter was used for excitation	j = 0.1 mA/cm^2^ at 1 V vs. RHE (20 times better than bare ZnO)	[73]
C_60_ (fullerene)/ZnO core–shell QDs	NaClO_4_ (0.5 M), I = 100 mW/cm^2^ from 300 W xenon lamp	j = 0.235 mA/cm^2^ at 1.23 V vs. RHE (6 times better than bare ZnO)	[74]
MLG/ZnO core nanoparticle	NaClO_4_ (0.5 M), I = 100 mW/cm^2^ from 300 W xenon lamp	j = 0.13 mA/cm^2^ at 0.2 V vs. Ag/AgCl (4.3 times better than bare ZnO)	[75]

## Data Availability

Data sharing is not applicable.

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
