# Peer review of "3D-ZnO Superstructure Decorated with Carbon-Based Material for Efficient Photoelectrochemical Water-Splitting under Visible-Light Irradiation"

_nanomaterials, 2023, doi:10.3390/nano13081380_

Round 1

Reviewer 1 Report

This minireview is a useful compilation. Its conception is appropriate, and the figures and tables suitably support the text. A theoretical question arose regarding Figure 5: What is the explanation for why the CB of MWCNT is lower than that of ZnO, while in the case of CQD (see (b) and (c)) the relation is reversed? (It has also been indicated in the attached manuscript decorated with highlights and remarks). 

The manuscript contains numerous errors in English grammar and style. Some of those are also indicated in the attached manuscript. (Various types of errors occur, e.g., singular vs. plural, usage of commas, sentences without verbs, typos, etc.).

Author Response

  1. This minireview is a useful compilation. Its conception is appropriate, and the figures and tables suitably support the text. A theoretical question arose regarding Figure 5: What is the explanation for why the CB of MWCNT is lower than that of ZnO, while in the case of CQD (see (b) and (c)) the relation is reversed? (It has also been indicated in the attached manuscript decorated with highlights and remarks).

    Response to Reviewer Comments:

    Thank you for your valuable feedback and comments. The reported result of MWCNT-ZnO material is the core–shell architecture, which limits the electron’s diffusion distance to the thickness of ZnO around the MWCNT. The bandgap of grown ZnO (in MWCNT) is 3.78 eV, while that of MWCNT is reported at 4.95 eV. A spontaneous electron transfer is therefore expected from ZnO to MWCNT, followed by the formation of a Schottky barrier, resulting in the promotion of a further flow of electrons from ZnO to CNT. The MWCNT plays the role of an electron acceptor, and the received electrons are promptly transported to the cathode via the external electrical circuit.

    On the other hand, the Cdots or CQDs were grown in the ZnO. This carbon material has a band gap of around 2.3 eV, and its conduction band and valence band edges are located at around −3.59 and −5.92 eV versus the vacuum level, respectively. While the ZnO bandgap is around 3.3 eV. Therefore, the band alignment of ZnO and Cdots or CQDs permits the effective passage of holes from the valence band of ZnO to HOMO of Cdots or CQDs and electrons from the LUMO of Cdots or CQDs to conduction band of ZnO.

  2. The manuscript contains numerous errors in English grammar and style. Some of those are also indicated in the attached manuscript. (Various types of errors occur, e.g., singular vs. plural, usage of commas, sentences without verbs, typos, etc.)

    Response to Reviewer Comments:

    Thank you for pointing out this issue. We have carefully revised and corrected some of the errors in the English grammar as you have indicated. Additionally, We have asked a professional editor to ensure the manuscript is free from grammatical or stylistic errors.

Reviewer 2 Report

The Authors prepared a manuscript related to synthesis of different morphology of ZnO. The content of Manuscript does not match with its title. Most parts of the manuscript is related to synthesis of ZnO and the observed morphologies. Also the part related to PEC does not written in details to present useful results to readers. 

I suggest rejection for this manuscript.

Author Response

1. The Authors prepared a manuscript related to synthesis of different morphology of ZnO. The content of Manuscript does not match with its title. Most parts of the manuscript is related to synthesis of ZnO and the observed morphologies. Also the part related to PEC does not written in details to present useful results to readers.

Response to reviewer comments:

Thank you for taking the time to review our manuscript. We appreciate your valuable feedback. We apologize for any confusion this may have caused. As for your information that the title is inspired by the latest issue updates about recent progress on the modification of 3D-ZnO superstructures by carbon-based material which was discussed in depth in section 4. Besides types of carbon-based materials, their effect on the PEC performance was also reviewed in this section.

Reviewer 3 Report

The manuscript presents an overview of the scientific results obtained over the past 11 years in the field of photo catalyst research based on three-dimensional ZnO superstructures intended for photo electrochemical water splitting.

The manuscript presents an overview of the scientific results obtained over the past 11 years in the field of photo catalyst research based on three-dimensional ZnO superstructures intended for photo electrochemical water splitting.

 3D ZnO − based super structures exhibit high solar collection efficiency, more reaction centers, large electron transfer, and low electron-hole recombination.

This review presents an analysis of the directions for the creation of various 3D − ZnO superstructures using synthesis methods and crystal growth modifiers, as well as their advantages and limitations. The prospects for using the modifications of materials with a carbon-based 3D − ZnO superstructure to increase the efficiency of water separation in the production of hydrogen are discussed.

The review is well structured, illustrated, contains an analysis of the results of 84 articles on the research topic.

The review will be interesting and useful for specialists, graduate students and students doing research in the field of photo electrochemical splitting of water under the action of visible light and creating the corresponding photo catalysts based on three-dimensional ZnO superstructures

Author Response

1. The manuscript presents an overview of the scientific results obtained over the past 11 years in the field of photocatalyst research based on three-dimensional ZnO superstructures intended for photoelectrochemical water splitting. 3D ZnO − based superstructures exhibit high solar collection efficiency, more reaction centers, large electron transfer, and low electron-hole recombination. This review presents an analysis of the directions for the creation of various 3D − ZnO superstructures using synthesis methods and crystal growth modifiers, as well as their advantages and limitations. The prospects for using the modifications of materials with a carbon-based 3D − ZnO superstructure to increase the efficiency of water separation in the production of hydrogen are discussed. The review is well structured and illustrated, and contains an analysis of the results of 84 articles on the research topic. The review will be interesting and useful for specialists, graduate students, and students doing research in the field of photoelectrochemical splitting of water under the action of visible light and creating the corresponding photocatalysts based on three-dimensional ZnO superstructures

Response to reviewer comments:

Thank you very much for your positive response. We are pleased to hear that you found this review well-structured and illustrated, and contains a comprehensive analysis of the results of 84 articles on the research topic.

We are glad that you found the manuscript interesting and useful for specialists, graduate students, and students doing research in the field of photoelectrochemical splitting of water under the action of visible light and creating the corresponding photocatalysts based on three-dimensional ZnO superstructures.

We appreciate your positive feedback on the overview of the scientific results obtained over the past 11 years in the field of photocatalyst research based on three-dimensional ZnO superstructures intended for photoelectrochemical water splitting. We are hopeful that this review will contribute to the advancement of the photoelectrochemical water-splitting field and the development of more efficient photocatalysts.
